



# Ocean signature of intense wind events in the Western Mediterranean Sea

Francesco Ragone[1,2], Andrea Meli[1], Anna Napoli[1], and Claudia Pasquero[1]

[1]Department of Earth and Environmental Sciences, University of Milan-Bicocca, Milan, Italy
[2]Laboratoire de Physique, ENS-Lyon, Lyon, France

**Correspondence:** claudia.pasquero@unimib.it

**Abstract.** The Western Mediterranean Sea is often subject to intense winds, especially during the winter season. The effects of the enhanced enthalpy and momentum fluxes on the upper ocean is investigated using sea surface temperature and sea surface height observational data products in the period 1993-2014. The maximum surface cooling associated with the anomalous ocean heat loss, with upwelling, and with diapycnal mixing is shown to occur a couple of days after the intense wind event, to be dependent on the wind intensity and to persist for over a month during winter, when deep water is formed, and for about 10 days during summer, when the thermocline is very shallow. The sea surface height signal reaches a minimum in correspondence of the intense wind, and normal conditions recover in about six weeks. Unlike for intense winds in the tropics, associated to tropical cyclones, no long term sea surface height anomaly is observed, indicating that the water column heat content is not significantly modified.

## 1 Introduction

Intense winds typically leave a cold wake on the surface of the ocean. The surface cooling is related to three different processes, their relative importance being dependent on the wind properties (Price, 1981; Mei and Pasquero, 2013). First, air-sea fluxes of sensible and latent heat, that are typically an enthalpy loss for the ocean, monotonically increase with wind speed. Second, the action of the winds induces a surface intensified current, whose vertical shear generates instability and produces vertical mixing with the underlying water, which is typically colder than the surface water. Third, when the winds are associated with a synoptic or mesoscale cyclonic pattern in the atmosphere, the generated Ekman flow leads to horizontal divergence at the surface and upwelling of deeper and colder water.

The overall net effect of the three processes is surface cooling. Studies on tropical regions have investigated the cold wake left at the ocean surface by tropical cyclones (for a systematic study see Mei and Pasquero, 2013), revealing that the amplitude of the cold surface anomaly depends on the intensity of the winds, on the translational speed of the atmospheric pattern causing them, and crucially on the stratification properties of the upper ocean (Mei et al., 2012; Mei and Pasquero, 2013). Furthermore, the SST recovery time scale has been estimated on the scale of a couple of weeks (Sanford et al., 2007; Price et al., 2008; Mei and Pasquero, 2013), as an effect of anomalous air-sea fluxes and mixed layer baroclinic instabilities associated with the front induced by the localized mixing (Boccaletti et al., 2007; Mei and Pasquero, 2012).



While the surface signature disappears within a few weeks, the subsurface warming associated with the wind intensified vertical mixing survives for a much longer time, resulting in a long term warming effect of tropical cyclones on the water column (Emanuel, 2001; Sriver and Huber, 2010; Jansen et al., 2010; Mei et al., 2013). As proposed in the cited works, the net long-term warming of the water column due to the action of the intense winds in the tropical regions can affect ocean heat

transport and impact processes that depend on ocean heat content. These studies have however been limited to the tropical regions, while little attention has been given to extra-tropical environments.

The aim of this work is to perform the same type of analysis in the Mediterranean Sea. This region is of special interest mainly because it is extremely peculiar in the sense that deep water formation occurs at the end of the winter season in its Western part, in particular in the Gulf of Lions, affecting the thermal properties of the water column, as documented for the

first time by the MEDOC experiment (MEDOC GROUP, 1970). The stratification of the Western Mediterranean shows in fact a strong seasonal variability. The Mediterranean mixed layer depth (MLD) seasonal variability is characterized by a basin scale deepening from November to February-March and an abrupt restratification in April, which is maintained throughout the summer and early autumn (D'Ortenzio et al., 2005; Houpert et al., 2014). Deep water formation is driven by particularly intense dry winds acting above the doming isopycnals generated by the cyclonic circulation present in the Gulf of Lyons, which

shallow the mixed layer and allow for the erosion of the upper ocean stratification related to the intrusion of warm and salty Levantine Intermediate Water (MEDOC GROUP, 1970; Schott et al., 1994). A systematic analysis of the short and long term effects of intense wind events on the properties of the Western Mediterranean Sea is lacking. With this work we perform this analysis exploiting high resolution observational datasets for surface winds, sea surface temperature and sea surface height.

The paper is structured as follows. In Section 2 we present the observational datasets and describe the statistical analysis we

have performed in order to identify the intense wind events and evaluate their impact on the state of the upper ocean. In Section 3 we discuss the results of this analysis. We first describe the short term (few weeks) behavior of sea surface temperature and sea surface height anomalies just after the occurrence of the intense wind event, both locally and area averaging over a region with the typical size of the dynamical structures responsible for the intense winds. We then study the persistence on seasonal scale of the signature that strong winds generate on the upper ocean. In Section 4 we draw our conclusions and discuss the

implications of our findings.

## 2   Materials and methods

### 2.1   Data

Daily sea surface temperature (SST) data for the period 1981 to November 2016 are obtained from the NOAA daily Optimum Interpolation Sea Surface Temperature dataset (OISST, https://www.ncdc.noaa.gov/oisst), an analysis combining observations

from different platforms (satellites, ships, buoys) interpolated on a regular global grid (Banzon et al., 2016) at 0.25° resolution. The methodology includes bias adjustment of satellite and ship observations (referenced to buoys) to compensate for platform differences and sensor biases. The version of OISST we consider uses as relevant satellite SST sensor the Advanced Very High





Resolution Radiometer (AVHRR), which, as an infrared instrument, can make observations at relatively high resolution but cannot see through clouds.

Daily mean sea height anomalies over the Mediterranean with a spatial resolution of 0.125° for the period 1993–2016 were obtained from the Copernicus Marine and Environment Monitoring Service (CMEMS, http://marine.copernicus.eu/) . They are
produced from the SL-TAC multi-mission altimeter data processing system. Optimal Interpolation is applied to all altimeter fields in order to obtain gridded sea level anomalies merging all the satellites data.

Sea surface wind data are taken from the V2.0 Cross-Calibrated Multi-Platform (CCMP, http://www.remss.com/measurements/ ccmp) gridded surface vector winds product (Atlas et al., 2011; Wentz et al., 2015). The V2.0 CCMP processing merges Remote Sensing System (RSS) radiometer wind speeds (SSM/I, SSMIS, AMSR, TMI, WindSat and GMI), QuikSCAT and ASCAT
scatterometer wind vectors, moored buoy wind data, and ERA-Interim model wind fields using a Variational Analysis Method (VAM, Hoffman et al. 2003) to produce 6 hourly maps of 0.25° degree gridded vector winds. VAM combines RSS instrument data with moored buoy measurements and a starting estimate of the wind field, provided by the ERA-Interim reanalysis winds. All wind observations (satellite and buoy) and model analysis fields are referenced to a height of 10 meters.

We have remapped the sea surface height data on the coarser 0.25° grid of the winds and sea surface temperature by bilinear
interpolation. The wind data have a time resolution of 6 hours, while sea surface temperature and height are provided daily. The wind dataset extends from July 1987 to July 2015, the sea surface temperature dataset from 1981 to November 2016, the sea surface height dataset from January 1993 to December 2016. We have therefore limited the analysis to the overlapping period 1993-2014.

The average wind intensity over the Mediterranean region computed from the CCMP dataset shows a strong non-stationarity
across the 1993-2014 period. The average kinetic energy of the winds increases by a 20% factor during this time period. This is likely to be a bias of the dataset, caused by inhomogeneities in the assimilation procedure used to merge the ERA-Interim first guess with observations (Ross Hoffman, personal communication). Due to the non stationary number of observational data, observations tend to weight more in the estimate towards the end of the time period, while in the first part the ERA-Interim background field dominates the average. Since ERA-Interim has a coarse spatial resolution, it tends to underestimate wind
intensity extremes with respect to observations. More intense wind events are thus observed in the second half of the time period rather than in the first half. Using the CCMP dataset to study climatic trends at global and regional scale (Zheng et al., 2016), at least in the Mediterranean area, is therefore questionable. This issue is of relatively little concern for our analysis though, as we are not interested in the temporal changes of the statistics of wind intensity. Having artificially weaker extreme winds in the first period of the dataset leads to miss events happening at earlier times, hence to have a poorer statistics overall,
but does not affect our results otherwise. See Appendix A for a more in depth discussion on this issue.





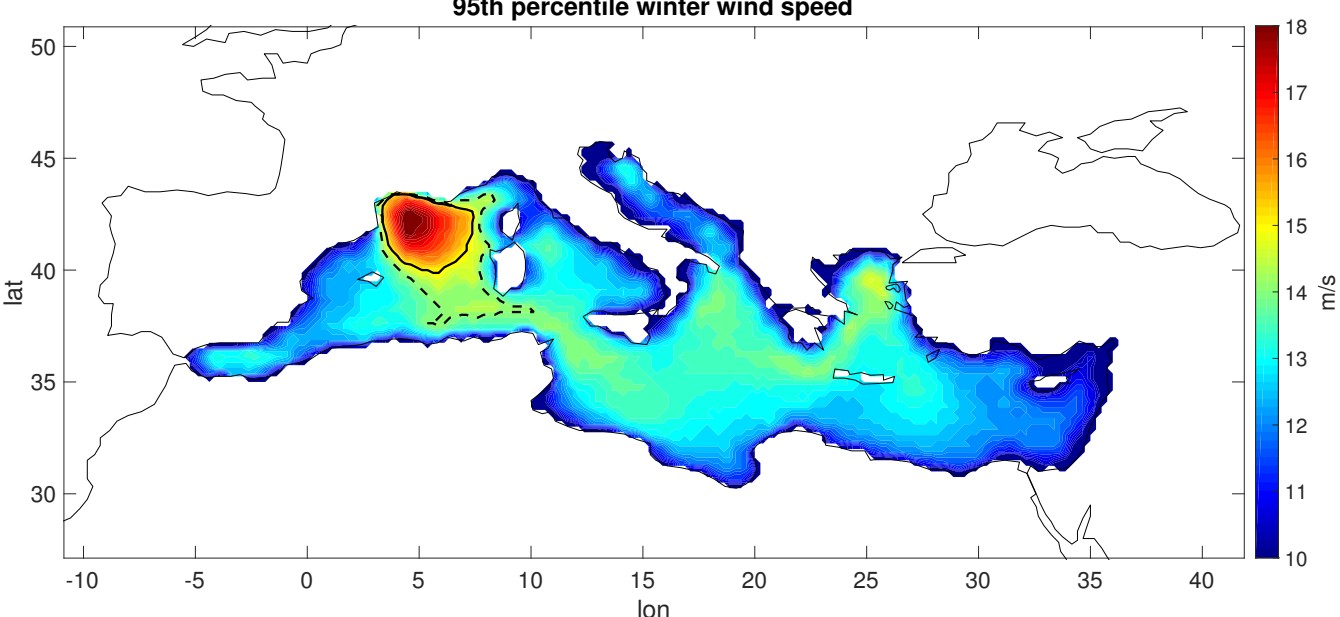

**Figure 1.** 95th percentile of 6-hourly winter wind intensity. The black solid line highlights the contour line at $15\ ms^{-1}$, enclosing the region in which the analysis presented in this study has been performed. The black dashed line highlights the contour line at $14\ ms^{-1}$, limited to the Western Mediterranean.

## 2.2 Methods

### 2.2.1 Intense wind events characterization

The intensity and frequency of winds in the Mediterranean region has a strong seasonal dependence. Strongest winds typically occur during the winter months, when the region is subject to the extratropical storms typical of the mid latitudes: about 90%
5  of the events within the upper fifth percentile in terms of wind intensity occur between November and March, and very few occur during the summer months from June to August. This aspect, together with the fact that the upper ocean conditions have a strong seasonal cycle which can lead to different ocean responses to intense winds in different seasons, is at the basis of the focus of the analysis presented in this paper to what in the following we call "winter", that is the 5 months period from November to March, and we include some summer (June, July, August) results for completeness only.
10    From the CCMP dataset we consider the instantaneous 6-hourly surface wind velocity vector $\overrightarrow{u}(\phi,\lambda,t_6)$ at latitude $\phi$, longitude $\lambda$ and time $t_6$. Figure 1 shows $u_{95}(\phi,\lambda)$, the 95th percentile of the local winter wind intensity $u(\phi,\lambda,t_6) = |\overrightarrow{u}(\phi,\lambda,t_6)|$. Locations with larger values of $u_{95}$ feature more intense and more frequent extreme wind events. Maximum values are found in the area of the Gulf of Lions in the Western Mediterranean, with secondary maxima present in the Eastern Mediterranean. We limit the analysis to the area characterized by $u_{95} > 15\ ms^{-1}$ (solid black line in figure 1), where winds intense enough





to induce a significant signal on the upper ocean state are relatively frequent. In this paper, we refer to this region as the Gulf of Lions. Note that considering a wider area by lowering the threshold to $u_{95} > 14\,ms^{-1}$ (dashed black line in figure 1) alters our results only very slightly from the quantitative point of view, and does not change them qualitatively, nor changes their statistical significance. In general we do not expect the results to change as long as the considered area remains in the Western

Mediterranean. Strong winds in the Eastern Mediterranean are due to distinctively different circulation patterns (Pfahl, 2014), and may yield to different results.

The area considered coincides with the area of deep water formation in the Western Mediterranean (D'Ortenzio et al., 2005; Houpert et al., 2014, 2016), where deep convection is in large part triggered exactly by the strong atmospheric forcing due to the presence of intense winds in the winter season (D'Ortenzio et al., 2005). Strong winds generate mixing that at the beginning

of the winter erodes the stratification, reducing the buoyancy of the subsurface warm and salty Levantine Intermediate Water. Underneath, a weakly stratified Western Mediterranean Deep Water favors deep convection that at the end of the winter period can reach the bottom.

The Gulf of Lions has a scale size of less than 500 km, and it is thus expected that spatial correlations (both in the atmospheric forcing and in the ocean conditions) are present. To avoid making statistics out of strongly correlated data, we define an intense

wind event as the occurrence $|\vec{u}(\phi, \lambda, t_6)| > a$, with $a$ a threshold varied between 16 $ms^{-1}$ and 20 $ms^{-1}$, for a temporal coordinate $(t_6)$ and for at least one couple of spatial coordinates $(\phi, \lambda)$ within the region inside the closed solid line in fig. 1. Furthermore, the wind dataset has a time resolution of 6 hours, while sea surface temperature and sea surface height have a time resolution of 1 day. For this reason, we define an intense wind day a day in which there is at least one intense wind event, as previously defined. Each intense wind day is assigned to the winter or summer dataset depending on its date of occurrence,

$t_i$, with $i = 1, .., N$, and with N=$N_w$, $N_s$ the total number of intense wind days in the winter and summer datasets, respectively.

### 2.2.2   Composite analysis of ocean signature of intense wind events

We study the impact of the intense wind days on the state of the upper ocean by means of a composite analysis. In the composite averages we mix values of the observable from different years and different calendar days, which are statistically not equivalent due to long term variations and seasonal cycle. Therefore, we remove from the sea surface temperature and the

sea surface height locally at each point $(\phi, \lambda)$ the long term linear trend and the seasonal cycle. We refer to the detrended and deseasoned sea surface temperature anomaly and sea surface height anomaly as SSTA and SSHA respectively.

Let us consider a generic surface observable $O(\phi, \lambda, t)$, in the following either SSTA or SSHA, with $t$ at daily resolution. We compute its average over the Gulf of Lions,

$$\overline{O}(t) = \frac{1}{\Omega} \int_{\Omega} O(\phi, \lambda, t) d\Omega. \tag{1}$$



and then consider its time evolution before and after the day of occurrence of an intense wind event, at a time lag $\tau$. We define the average composite evolution of the area averaged observable at a time lag $\tau$ from the intense wind day as

$$\overline{O}_{w,s}(\tau) = \frac{1}{N_{w,s}\Omega} \sum_{i=1}^{N_{w,s}} \int_{\Omega} O(\phi, \lambda, t_i + \tau) d\Omega. \tag{2}$$

where the subscript $w, s$ refers to the winter or summer occurrence of the event considered. The composite evolution for sea

surface temperature and sea surface height anomalies have been computed for values of $\tau$ ranging from $-90$ days to $+180$ days. Note that with this definition of the events, days when local intense wind events occur in one hundred grid points weight in the sum as much as days when local intense wind events occur in ten grid points.

The error on the estimates given by equation 2 is computed as the standard deviation related to the sum involved in the formulas, divided by the square root of $N_{w,s}$, the number of days in the considered region when at least one intense wind event

has been recorded.

## 3   Results

### 3.1   Extreme winds record

The number of local intense winds in the winter period (November through March), decreases from about 140,000 for the lowest value of the threshold, $a=16\ m\,s^{-1}$, to less than 28,000 for the largest value, $a=20\ m\,s^{-1}$. Correspondingly, the number

of days featuring at least one intense event $N_w$ decreases from 990 to 342. Values of $N_w$ of a few hundreds allow for a robust statistical analysis. During one day, the fraction of area on average occupied by winds locally above the threshold decreases from about 78% for the lowest threshold to about 44 % for the threshold at 20 $m\,s^{-1}$. The average number of days per year featuring intense winds similarly decreases from about 45 to about 15 over the winter months. These events are thus relatively common for this area; however, they are extremely rare when considering the entire Western Mediterranean basin.

During summer (June through August) the intense winds events are significantly rarer. In the considered area the number of local intense winds goes from about 8,200 for $a=16\ m\,s^{-1}$, to just 176 for $a=20\ m\,s^{-1}$. These values correspond to 122 and 11 days featuring at least one intense event. Therefore, statistical robustness is expected only for low values of the threshold. The area covered on average by intense winds every day recorded as intense wind day is also substantially smaller, dropping form 38% for $a=16\ m\,s^{-1}$ to just 9% for $a=20\ m\,s^{-1}$. The average number of intense wind days per year similarly drops from about

6 to about 0.5. Despite these numbers being so low, even in summer the Gulf of Lions is still the area of the Mediterranean basin with the strongest winds. Two other areas of characterized by relatively strong wind intensity are present, in the Alboran sea and in the Aegean sea. With respect to the winter case, in summer they are relatively closer to the intensities found in the Gulf of Lyons, which nevertheless remains the dominant region of intense winds activity in the Mediterranean basin.





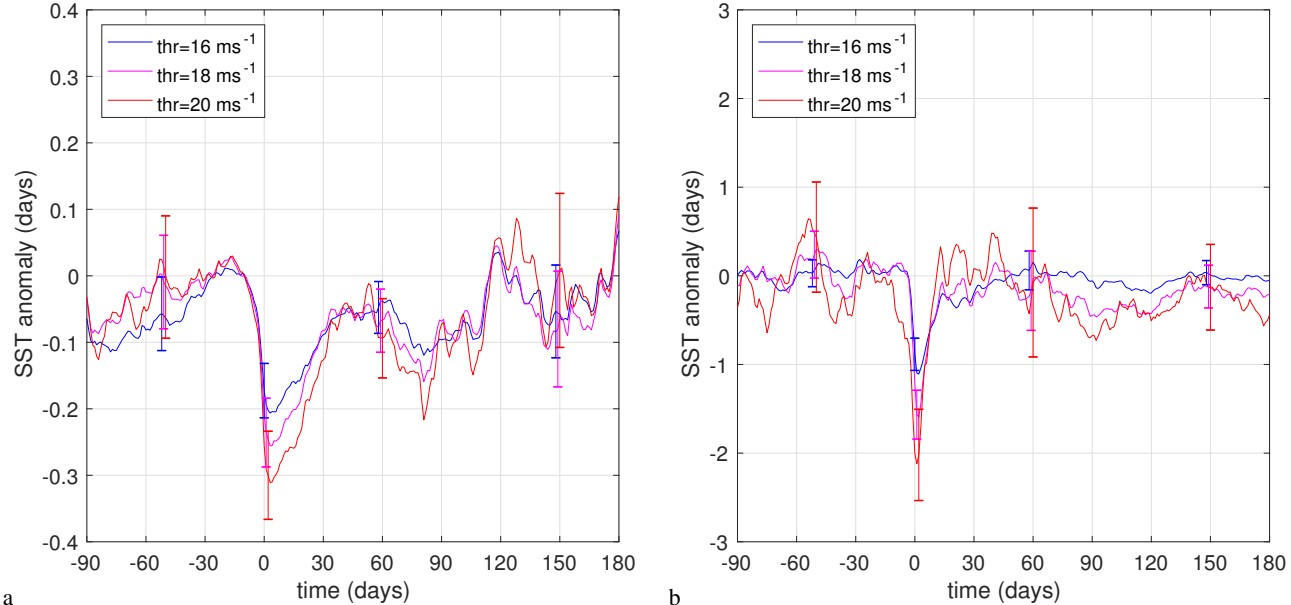

**Figure 2.** Composite average of time evolution before and after the intense wind events of area averaged SSTA for (a) winter and (b) summer seasons, for wind speed thresholds of 16 (blue), 18 (magenta) and 20 (red) $ms^{-1}$. Error bars around -50, 5, 60 and 150 days are computed as 2 standard deviations of the composite sample. Note the different axis range in the two panels.

### 3.2 Composite analysis of SSTA

The composite average of the time evolution of winter SSTA before and after the occurrence of intense wind events, computed for the different values of the threshold is shown in figure 2a. The area averaged sea surface temperature anomaly drops by $O(10^{-1})\,K$, reaching a minimum a few days after the occurrence of the intense wind. Note that the drop actually starts slightly

5     before the occurrence of the intense wind event. This signal might be due to the fact that SST are provided daily in the OISST dataset, but they have been obtained by performing a bias adjustment using a spatially smoothed 7-day in situ SST average, which might anticipate the temporal signal (Reynolds et al., 2007). Additionally, weather conditions favoring surface cooling are probably present in the days preceding the individual days with intense winds as those are usually embedded in a synoptic scale perturbation, and during wintertime winds have a temporal correlation of about a week (not shown).

10     The cooling in correspondence of the anomalous winds can be in principle related to three mechanisms: the enhanced enthalpy fluxes associated with intense winds, the mixing of the upper ocean generated by vertical shear instabilities of horizontal currents induced by the winds at the surface, and the Ekman upwelling induced by the cyclonic winds. Each of those processes is dependent on the intensity of the winds, with stronger winds generating a larger cooling. In fact, data in fig. 2a show this sensitivity, with a minimum of about -0.2 K for winds stronger than 16 m/s and a minimum of about -0.3 K for winds stronger

15     than 20 m/s.





The values of the SST anomalies are very small, consistent with the fact that the deep water formation region in the Gulf of Lions is usually very weakly stratified during the winter season. Strong winds bring up deep water, but this is only marginally colder than the water above. Similarly, enhanced enthalpy fluxes associated with intense winds have a minimal impact on the SST reduction when the loss is shared among the whole water column and not limited to a surface layer.

After reaching a minimum a few days after the occurrence of the strong wind, SSTA slowly returns to zero. Two time scales seem to be involved. A first roughly exponential recovery is followed by a series of oscillations. SSTA returns to a statistically stable state after about 100-120 days. The relaxation time scale of the first recovery is about 40 days, computed by fitting with an exponential the SSTA composite average in the first 60 days after the maximum of the drop, and does not depend on the value of the threshold. The first recovery period lasts for about 60 days, at the end of which the system has not yet properly
reached a state that is zero compatibly with the error bars. A subsequent oscillation on a period of about 20 days follows, even if the extremely noisy nature of the signal makes it difficult to interpret its behavior after the first 60 days.

    For completeness, we report in fig. 2b also the results of the analysis relative to the summer months (June through August). The signal is noisier than for the winter case due to the limited number of events but it is clear that the maximum SST anomaly is one order of magnitude larger than for winter events, reaching values of about 2.5K for the strongest winds (note
the different scale in the temperature axis). Clearly, the strong stratification typical of summer months (D'Ortenzio et al., 2005) is responsible for the large drop in SST as the shallow strong stratification favors the upwelling of water several degrees colder than the surface water. The initial recovery time scale is about ten days, much shorter than during winter, probably because the large short wave radiation input quickly re-stratifies the water column. It remains to be determined whether a warm anomaly survives below the surface for longer times, as it happens for tropical cyclones.

## 3.3   Composite analysis of SSHA

Sea surface height is influenced by both dynamical and thermodynamical processes related to the occurrence of the strong wind. The inverse barometer effect associated with the low pressure system that typically embeds these events can be responsible for a small increase in the sea level. Ekman divergence and upwelling characteristic of the cyclonic wind circulation causes a drop in SSH. Heat loss at the air-sea interface further reduces SSH because of thermal contraction of the water column. It is unclear
at present what is the magnitude of each of these mechanisms.

    The composite averaged winter and summer SSHA from 90 days before the event to 180 days after are shown in figures 3a and 3b, respectively. The SSHA signal is less noisy on short time scales of one or few days with respect to the SSTA signal. This is likely due to the fact that SSHA is a vertically integrated measure which depends on the thermal properties of the entire water column, and it is thus less sensitive to the fast fluctuations of the surface fluxes due to weather activity. Beside that, the
general behavior of the SSHA signal qualitatively resembles that of the SSTA signal, with some notable differences.

    Composite SSHA starts decreasing well before, 10 to 15 days, the occurrence of the strong wind event, much earlier than the SSTA signal. This is likely not related to the passage of the atmospheric dynamical structure itself, and it is unlikely that it is due to some interpolation procedure in the creation of the dataset. The same feature has been observed in a similar analysis



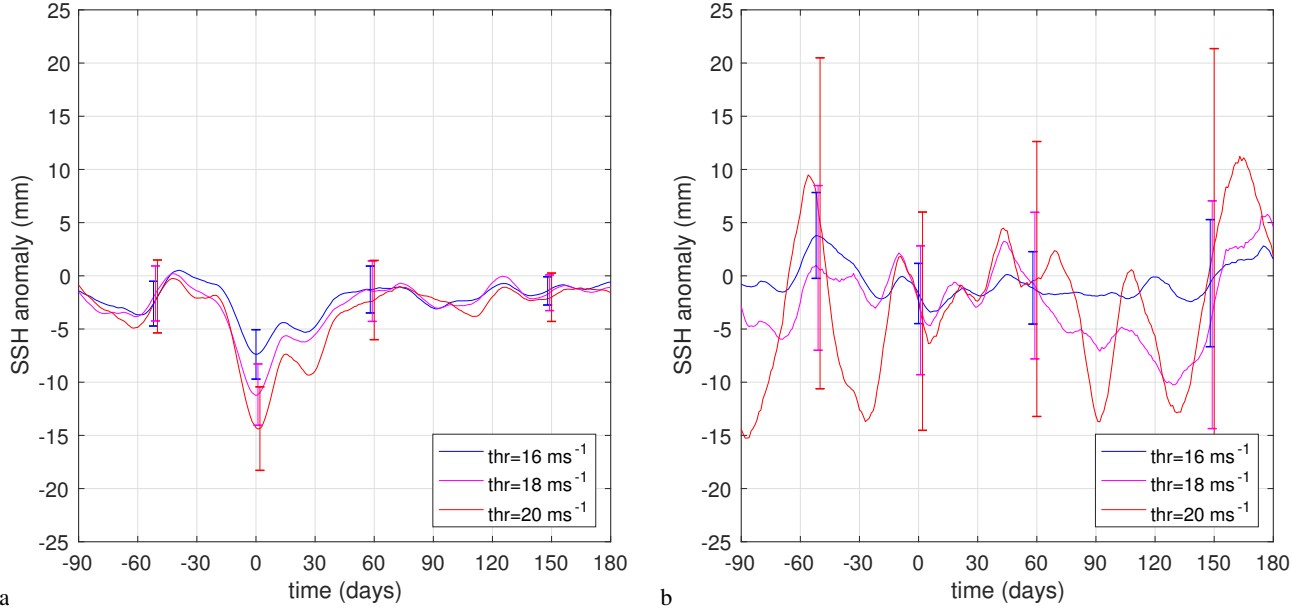

**Figure 3.** Composite average of SSHA time evolution before and after the intense wind events for (a) winter and (b) summer seasons, for wind speed thresholds of 16 (blue), 18 (magenta) and 20 (red) $ms^{-1}$. Error bars at -50, 5, 60, and 120 days are computed as 2 standard deviations of the composite sample.

of the effect on SSHA of the passage of hurricanes in the Tropics (Supplementary Information in Mei et al. (2013)), but to our knowledge no explanation has been given yet.

The SSHA signal shows a sensitivity to the intensity of the wind, with an area averaged drop peaking at -12 mm for winds stronger than 16 m/s and at -17 mm for winds stronger than 20 m/s. The minimum SSHA is reached exactly at the occurrence of the intense wind events, while for SSTA it was a few days later. Clearly this difference might be in part due do the interpolation procedure to obtain spatially and temporally uniform fields, but the physical mechanism at the base of the signal can also be relevant: SSH responds to enthalpy fluxes and Ekman pumping, which impact the sea level on short timescales, while the signal in SST requires mixing that develops by shear instabilities and convection, which require time to develop.

Winter SSHA shows a similar initial recovery period as SSTA: it increases for the first 60 days, and is significantly different from zero, according to the uncertainty estimate, at least for the first 40 days. Thanks to the fact that the SSHA is less noisy than SSTA, it is more clear the appearance of an oscillation on a period of about 20 days, whose nature remains unclear, but whose presence appears ubiquitous in our composite analysis.

During the summer season, no significant SSHA is detected, due to the large fluctuations associated to the low number of events occurring during this season (see fig. 3b). It might also be that the signal is smaller during summer than winter because events are not associated to synoptic scale cyclones and the Ekman divergence is smaller.



## 4  Conclusions

We have performed a composite analysis of the impact of intense wind events on the state of the upper ocean in the Western Mediterranean area from 1993 to 2014, using publicly available high resolution observational datasets for wind, sea surface temperature and sea surface height. In this region the vast majority of intense wind events occur in winter in the Gulf of Lions,

where deep water formation occurs (D'Ortenzio et al., 2005; Houpert et al., 2014, 2016).

Strong winds at the surface induce mixing of the upper ocean water, which results in a drop of sea surface temperature in the days immediately following the events. During winter time, the maximum drop is only of the order of $0.1^{o}$C, as the water column is well mixed, while during summer time the drop is one order of magnitude larger due to the strong stratification and the shallow mixed layer. More intense winds in general induce deeper mixing, and therefore bring up to the surface deeper,

usually colder, water. The sea surface temperature anomaly drop in fact increases with the threshold set to select the intense wind events. The sea surface temperature anomaly goes back to zero after about 40-60 days. This recovery time, estimated with an exponential fit, does not significantly depend on the wind intensity.

We have also analyzed sea surface height. The response of sea surface height anomalies is characterized by a drop of a few cm, caused by divergent Ekman flow induced by the cyclonic wind stress during the passage of the storm and also to

the evaporative enthalpy loss (Mei et al., 2013). The signal is particularly evident during winter, when the cyclonic nature of the synoptic scale perturbation in which the strong winds are embedded is responsible for a large Ekman divergence. The amplitude of the SSHA depends on the wind intensity. The recovery time is consistent with what found for sea surface temperature anomalies, and does not depend on the wind intensity. Unlike for the tropics, where intense winds associated with tropical cyclones are responsible for long term warming of the upper ocean, no long term SSHA signal is observed. This is

probably related to the very different stratification of the water column: In the tropics a warm mixed layer is present at the surface and intense winds mix water at the thermocline, bringing warm water down which, after the upper layer returns to normal conditions, is responsible for a net increase of the water column heat content (Mei and Pasquero, 2013). In the Gulf of Lyons, on the other hand, the very weak winter stratification prevents the burial of heat in the lower layers as the mixing water is nearly homogeneous in temperature. Thus the SSHA returns to zero after a few weeks.

It would be interesting to further characterize the upper ocean response to intense winds in terms of the salinity. This will become possible in the next future, when satellite missions will have collected several years of sea surface salinity data.

The impact of the strong winds during a storm on the state of the upper ocean can lead to memory effects influencing the coupled atmosphere-ocean system at different time scales. In the Tropics, this extends to seasonal time scales: it has been shown that hurricanes on the long term pump heat into the ocean leading to a warming detectable 4-6 months after the event (Emanuel,

2001; Sriver and Huber, 2010; Jansen et al., 2010; Mei et al., 2013). This work shows that in the Western Mediterranean this process is not at play,

Outside the Tropics, and specifically in the Western Mediterranean area, it has been shown in a modeling case study (Lebeaupin Brossier et al., 2013) that a strong wind event can influence weather on a time scale of a week through the signature on the SST and its consequent impact on the air-sea fluxes. Similarly, air-sea coupling on sub-monthly time scale has





been found to be important in the modulation of heavy precipitation events in this and other regions of the Mediterranean in a climatic modeling study (Berthou et al., 2016).

Our work confirms from observational data that the signature of intense wind events on the state of the upper ocean in the region persists for weeks, and has the potential to influence air-sea processes on different time scales. The fact that surface

thermal anomalies persist for at least 30 days, confirms thus the potential of (sub-)monthly scale feedbacks.

## Appendix A:  Appendix - Spurious trends in CCMP wind data in the Mediterranean region

The statistical properties of wind intensity over the Mediterranean region in the CCMP dataset show a strong non stationarity during the period we have analyzed. In particular, the average kinetic energy of the winds over the basin increases by a 20% factor, from about 18 $m^2 s^{-2}$ to about 22 $m^2 s^{-2}$ (red line of figure A1). In the previous version of the CCMP product (version

1.1, light blue) the trend is even stronger, as the estimate at later times is close to the estimate of the new product, while they differ at the beginning of the time period. However, such signal appears not to be present in other observational dataset that are part of the assimilation process of the CCMP product.

We have considered the same annual average of KE in the ERA-Interim reanalysis (which is used as first guess or background field for the CCMP estimate), in the SSM/I radiometer data (http://www.remss.com/missions/ssmi/) and in the ASCAT

scatterometer data (http://www.remss.com/missions/ascat/). Additionally, we have compared these results with a 30 years long run of the atmospheric regional model WRF in non-hydrostatic setup, at 4 Km horizontal resolution, forced with ERA-Interim data (Pieri et al., 2015).

Average KE appears to be stationary in ERA-Interim (black) with values very similar to the CCMP estimate at the beginning of the time period. No significant trends are present in the SSM/I radiometer data (blue) and ASCAT scatterometer data

(magenta), both of which feature larger values than ERA-Interim, closer to the CCMP estimate towards the end of the time period. Note that there is no trend also in a 30 years long run with the high-resolution atmospheric regional model WRF (green) forced with ERA-Interim data. WRF estimate features the largest values of the group, close to the ASCAT estimate.

Summarizing, the CCMP estimate is almost identical to the low values of the ERA-Interim first guess at the beginning of the time period, and then increases to larger values typical of the observational datasets or high-resolution numerical results by

the end of the time period. This is an artificial behavior, caused by the increase in the number of satellite observations and the relative offset in mean speed between the satellite observations and ECMWF (Ross Hoffman, personal communication). As a result the reanalysis background field weights more on average at the beginning of the assimilated period, while observations gradually become more important at later times. The background field given by the reanalysis has a relatively coarse spatial resolution of about 80 Km, while observational data have a resolution of about 25 Km and the WRF run of about 4 Km.

Therefore, the background field underestimates the largest values of the the wind speed $|\vec{u}|$, which, being KE a quadratic function of $|\vec{u}|$, leads to a severe underestimate of KE. This ultimately leads in the CCMP product to a lower estimate of KE at earlier times, when the contribution of the background field is dominant.



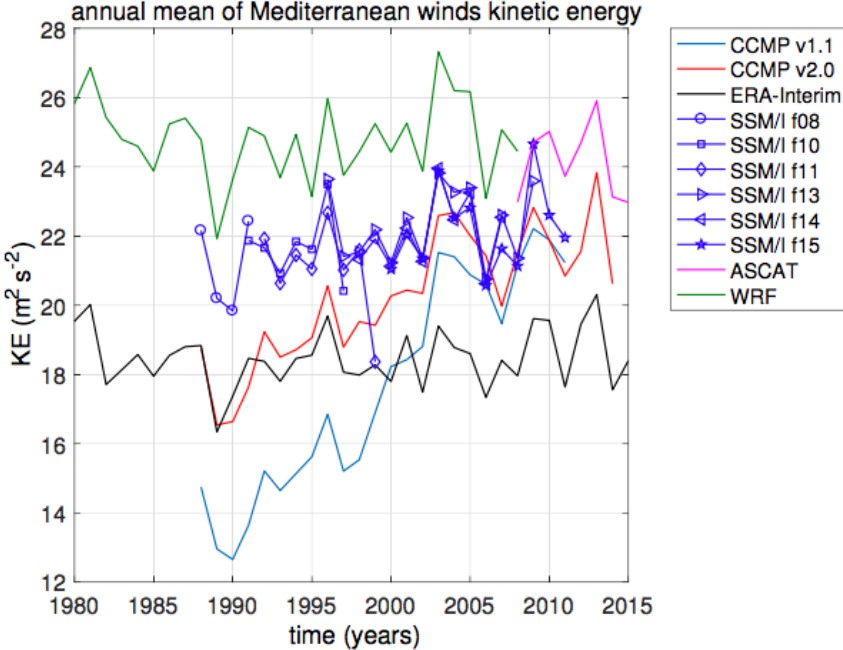

**Figure A1.** Annual mean of specific kinetic energy of surface winds averaged over the Mediterranean basin, from different sources. We compare results from the CCMP v1.1 (light blue) and v2.0 (red) products against ERA-Interim (black) data, radiometer (blue, for different instrument each covering a different time period) and scatterometer (magenta) data. We present also the results obtained by a high resolution 30 years long run with WRF in non-hydrostatic setup and forced by ERA-Interim data (green).

The presence of this spurious trend makes questionable to use the CCMP dataset for studying global and regional trends, at least in the Mediterranean region, despite having be used sometimes for this purpose in the recent past (e.g. Zheng et al., 2016). For our study however this is not a serious issue, as we are not interested in the statistics of wind speed by itself. The underestimate of strong winds at earlier times results in our case simply in missing intense wind events in the first part of the dataset (indeed we observe that about two thirds of intense wind events are found the second half of the considered period). Since we study the response of the state of the upper ocean to intense wind events, and since we can safely assume that the nature of this interaction has not changed with time, we only have a problem of loss of statistics rather than of biasing.

*Competing interests.* The authors declare no competing interests.

*Acknowledgements.* This article is an outcome of Project MIUR – Dipartimenti di Eccellenza 2018-2022. CP acknowledges support from the Flagship Project RITMARE funded by the Italian Ministry of Education, University and Research.



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
