# Peer review of "Ocean signature of intense wind events in the Western Mediterranean Sea"

_Ocean Science, 2018_

## Referee Comment (RC1) · Anonymous Referee #1 · 16 Oct 2018

The paper presents a statistical evaluation of the ocean response to strong winds in the Western Mediterranean region. In this study, the authors use composite products of SST, SSH and of wind fields to identify the ocean response(s) to strong wind and evaluate the duration of the anomalies formed. The methodology is correctly described and the results (only composite averages of SST and SSH) appear quite convincing, even if physical explanations are often mentioned but are not really documented.

My main concern is about the data used, in particular for the wind field with the CCMP product that has a space/time resolution of 0.25°/6hrs. It seems that it can miss some very fine/short events that are frequent in the Mediterranean basin (cyclones, medicanes, thunderstorms, or bora events for example). In the annexe, it appears that a downscaling of the ERA-interim reanalyses done with WRF at a 4km-resolution is

available and shows larger values of wind speed compared to CCMP. The WRF run also has the advantage of being homogeneous in time. So, it would be very useful to add a discussion about the issues linked to the use of CCMP (not only trend but also underestimation!) and the potential impact on the results (notably for what concerns the validity of the thresholds chosen).

I also suggest the authors to revise the paper when speaking about deep water formation (notably the introduction) as, first, it is not enough mentioned that it is an intermittent process with a large interannual variability (Somot et al. 2018). In particular, the ocean vertical stratification is a key factor for this interannual variability. Moreover, very recent results moderate now the direct role of the wind on deep water formation (Giordani et al. 2017, Waldman et al. 2018).

Considering these two main points, I suggest the authors to revise the paper before publication, in particular the introduction and the conclusion where discussions could take place. In my opinion, the abstract should also be revised, because the results shown here can not clearly permit to relate the upper ocean response to any enthalpy flux (heat loss) or diapycnal mixing. This also goes back to my comment about, in general, a lack of documentation of the ocean processes involved.

- - -

Other comments:

- p1, line 2 (abstract): 'The effects . . . are...'

- p4, line 14: 'winds are intense enough'

- p5, line 8: delete 'exactly'

- p5, equation 1: please, explain $\Omega$

- - -

Giordani, H., C. Lebeaupin-Brossier, F. Léger, and G. Caniaux (2017): A PV-approach

for dense water formation along fronts: Application to the Northwestern Mediterranean, J. Geophys. Res. Oceans, 122, 995–1015, https://doi.org/10.1002/2016JC012019.

Somot, S., Houpert, L., Sevault, F. et al. (2018): Characterizing, modelling and understanding the climate variability of the deep water formation in the North-Western Mediterranean Sea. Clim. Dyn., 51: 1179. https://doi.org/10.1007/s00382-016-3295-0

Waldman, R., Brüggemann, N., Bosse, A., Spall, M., Somot, S., and Sevault, F. (2018): Overturning the Mediterranean thermohaline circulation. Geophysical Research Letters, 45, 8407–8415. https://doi.org/10.1029/2018GL078502

---

## Referee Comment (RC2) · Anonymous Referee #2 · 22 Oct 2018

In this paper, the authors evaluate the signature of intense wind events in the North-western Mediterranean Sea (NWMS) on sea surface temperature (SST) and sea surface height (SSH). For that, they use available datasets of sea surface wind (CCMP), SST (OISST) and SSH anomalies (CMEMS). Focusing their study over the winter period and over the Gulf of Lions (but also showing summer results), they compute the evolution of SST and SSH anomalies during the period preceding and following intense wind events to estimate the amplitude and duration of the signature of those events on the SST and SSH. They find that in winter, winter wind events induce a O(0.1K) cooling of the surface and a few cm drop of SSH that take about 40-60 days to disappear.

***General comment***

While it is not uninteresting, this finding itself is not enough to be published without

further analysis, and this work needs significant improvement to represent a valuable contribution to our knowledge of the air-sea interactions in the NWMS. The most problematic points is the total absence of physical explanations. Moreover, the authors seem to completely ignore most of previous and recent studies that were done and published on this topic. They would greatly increase their knowledge and understanding of the functioning of the air-sea interactions in the Mediterranean Sea and NWMS by reading those studies, which would actually help them to propose some physical explanations for their finding. I provide a non-exhaustive list of suggested readings at the end of this review. In particular, winter ocean convection is a key process in the dynamics of winter surface characteristics in the region. To improve the quality of their physical interpretation, I strongly suggest that the authors contact teams that develop and provide numerical modeling tools of the atmosphere and ocean circulations in this region (e.g. the groups of Climate Modelling Laboratory and Impacts at ENEA, of CNRM at Météo-France/CNRS, groups involved in MED-CORDEX...) : using model outputs would allow them to disentangle the contributions of different factors and fluxes (surface forcing, lateral advection, vertical mixing, etc) to the surface cooling and drop observed here. In the following I provide some more detailed comments that could help the authors improve their analysis.

***Specific comments***

- As stated by the authors, the NWMS present some peculiar features compared to other regions of the world ocean. First this region is one of the few regions in the world ocean where deep convection occurs. Due to strong vertical mixing that occurs regularly throughout the whole water column in this region, the temperature, salinity and density vertical profiles are very weakly stratified, in particular the temperature does not go below ∼13°C. Consequently, when cold winter wind events induce a cooling, hence a density increase, of the surface layer, the water is vertically mixed with this relatively warm underlying water, and the SST can not go below ∼13°C (see e.g. Waldman et al., 2017). This explain the "small" SST anomaly induced by wind events observed by

the authors in winter compared to the summer. Moreover, the authors mention the contribution of enthalpy loss, vertical mixing induced by vertical shear of currents, Ekman pumping to the sea surface temperature anomaly during winter wind event. One essential factor is also this vertical mixing. The authors should try to provide a quantitative estimate of the contribution of those factors to the finding observed in their analysis. Moreover, many studies (see list below) have examined the respective contributions of vertical vs. horizontal, in particular mesoscale features, in the deep convection and following restratification processes : using a model, and in-situ data available over a long period (e.g. at the LION buoy mooring) could help the authors to perform fluxes analysis identify the contribution of surface, lateral and vertical fluxes to the SST cooling then recovery observed in their analysis. Similarly, model outputs would help to understand the dynamics involved in the signature on SSH.

- Winter, where deep convection occurs, and summer dynamics are very different, so they should be evaluated and physically explained separately.

- The authors perform their analysis over an area that they call "Gulf of Lions", that actually gather both the Gulf of Lions shelf (depth < 200m) and the neighboring open sea (where the depth can exceed 2000m). Both regions show different behaviors : over the shelf, winter cooling acts over a very shallow column, that is quickly mixed and that can therefore be cooled much more than the open ocean column, where the vertical mixing with relatively warm water over a much deeper column prevents the surface to become really cold ($\sim$13°C). Both areas should be analysed separately.

- In the Mediterranean sea, due to low stratification hence low Rossby deformation radius, the spatial scales of mesoscale processes is very small (of the order of 1-10 km). CMEMS SSH data are gridded products that can not capture those small scale structures. The authors should be aware about this limitation of using gridded data products to estimate the signature of wind events on the SSH (see for example Herrmann et al., 2009; Herrmann et al. 2017).

- The authors acknowledge the fact that the CCMP dataset does not represent wind consistently over the 1993-2014 period. Though they argue that this should not hinder their statistically analysis which is based on intense event only, this assumption should be verified, e.g. by examining the same variables in a coupled model that does not involve any data assimilation.

Some publications that the authors should consult (non exhaustive list, of course the authors will find many other interesting references in the bibliography lists of those papers):

Artale et al. (2010). An Atmosphere-Ocean Regional Climate Model for the Mediterranean area: Assessment of a Present Climate Simulation. Climate Dynamics doi:10.107/s00382-009-0691-8

Béranger et al. (2010). Impact of the spatial distribution of the atmospheric forcing on water mass formation in the Mediterranean Sea. J. Geophys. Res., 115, C12041. doi : 10.1029/2009JC005648

Durrieu de Madron et al. (2013). Interaction of dense shelf water cascading and open-sea convection in the northwestern Mediterranean during winter 2012. Geophys. Res. Lett., 40 :1379–1385. doi : 10.1002/grl.50331

Estournel (2016). High resolution modelling of dense water formation in the north-western mediterranean during winter 2012-2013 : Processes and budget. Journal of Geophysical Research : Oceans, 121(7) :5367–5392, 2016. doi : 10.1002/2016JC011935.

Herrmann and Somot (2008). Relevance of ERA40 dynamical downscaling for modeling deep convection in the North-Western Mediterranean Sea. Geophys. Res. Let., 35, L04607, http://dx.doi.org/10.1029/2007GL032442

Herrmann et al. (2008). Modeling deep convection in the Northwestern Mediterranean Sea using an eddy-permitting and an eddy-resolving model: case study of winter 1986-

87. J. Geophys. Res. 113, C04011, http://dx.doi.org/10.1029/2006JC003991

Herrmann et al. (2009). Monitoring open-ocean deep convection from space. Geophys. Res. Let., 36, L03606, http://dx.doi.org/10.1029/2008GL036422, See also Nature Research Highlights, Vol 457, 26 February 2009

Herrmann et al. (2010). What induced the exceptional 2005 convection event in the northwestern Mediterranean basin? Answers from a modeling study. J. Geophys. Res.,115, C12051, doi:10.1029/2010JC006162

Herrmann et al. (2011). Representation of wind variability and intense wind events over the Mediterranean sea using dynamical downscaling: impact of the regional climate model configuration. NHESS, 11, 1983-2001, 2011, doi:10.5194/nhess-11-1983-2011

Herrmann et al. (2017), Long-term monitoring of ocean deep convection using multisensors altimetry and ocean color satellite data. J. Geophys. Res. Oceans. doi:10.1002/2016JC011833

L'Hévéder et al. (2013). Interannual variability of deep convection in the northwestern mediterranean simulated with a coupled aorcm. Climate Dynamics, 41(3-4) :937–960. doi : 10.1007/s00382-012-1527-5.

Madec et al. (1991). A three-dimensional numerical study of deep-water formation in the Northwestern Mediterranean Sea. J. Phys. Oceanogr., 21(9) : 1349–1371.

Schroeder et al (2006). Deep and intermediate water in the western Mediterranean under the influence of the Eastern Mediterranean Transient. Geophys. Res. Lett., 33, L21607. doi : 10.1029/2006GL027121.

Schroeder et al (2008). An extensive western Mediterranean deep water renewal between 2004 and 2006. Geophys. Res. Lett., 35, L18605. doi : 10.1029/2008GL035146.

Schroeder et al. (2010). Abrupt warming and salting of the Western Mediterranean

Deep Water: atmospheric forcings and lateral advection. J. Geophys. Res., 115, C08029, doi:10.1029/2009JC005749

Somot et al. (2018). Characterizing, modelling and understanding the climate variability of the deep water formation in the North-Western Mediterranean Sea. Climate Dynamics. doi: 10.1007/s00382-016-3295-0

Waldman et al. (2017), Modeling the intense 2012-2013 dense water formation event in the northwestern Mediterranean Sea: Evaluation with an ensemble simulation approach. J. Geophys. Res. Oceans. doi:10.1002/2016JC012437

Waldman et al. (2016). Estimating dense water volume and its evolution for the year 2012-2013 in the North-western Mediterranean Sea: An observing system simulation experiment approach. J. Geophys. Res., doi:10.1002/2016JC011694